# Sulforaphane Attenuates Neutrophil ROS Production, MPO Degranulation and Phagocytosis, but Does Not Affect NET Formation Ex Vivo and In Vitro

**DOI:** 10.3390/ijms24108479

**Published:** 2023-05-09

**Authors:** Shiori Wakasugi-Onogi, Sihui Ma, Ruheea Taskin Ruhee, Yishan Tong, Yasuhiro Seki, Katsuhiko Suzuki

**Affiliations:** 1Graduate School of Sport Sciences, Waseda University, Tokorozawa 359-1192, Japan; 2Health Nutrition, Graduate School of Agricultural and Life Sciences, The University of Tokyo, Tokyo 113-8657, Japan; 3Research Fellow of Japan Society for the Promotion of Sciences, Tokyo 102-0083, Japan; 4Faculty of Sport Sciences, Waseda University, Tokorozawa 359-1192, Japan

**Keywords:** sulforaphane, whole blood, neutrophil, ROS, inflammation, degranulation, phagocytosis, NETs formation

## Abstract

Sulforaphane has several effects on the human body, including anti-inflammation, antioxidation, antimicrobial and anti-obesity effects. In this study, we examined the effect of sulforaphane on several neutrophil functions: reactive oxygen species (ROS) production, degranulation, phagocytosis, and neutrophil extracellular trap (NET) formation. We also examined the direct antioxidant effect of sulforaphane. First, we measured neutrophil ROS production induced by zymosan in whole blood in the presence of 0 to 560 µM sulforaphane. Second, we examined the direct antioxidant activity of sulforaphane using a HOCl removal test. In addition, inflammation-related proteins, including an azurophilic granule component, were measured by collecting supernatants following ROS measurements. Finally, neutrophils were isolated from blood, and phagocytosis and NET formation were measured. Sulforaphane reduced neutrophil ROS production in a concentration-dependent manner. The ability of sulforaphane to remove HOCl is stronger than that of ascorbic acid. Sulforaphane at 280 µM significantly reduced the release of myeloperoxidase from azurophilic granules, as well as that of the inflammatory cytokines TNF-α and IL-6. Sulforaphane also suppressed phagocytosis but did not affect NET formation. These results suggest that sulforaphane attenuates neutrophil ROS production, degranulation, and phagocytosis, but does not affect NET formation. Moreover, sulforaphane directly removes ROS, including HOCl.

## 1. Introduction

Inflammation is a fundamental process for removing pathogens or dead cells from damaged tissues and maintains homeostasis [1]. The inflammatory response necessitates the production of reactive oxygen species (ROS) and cytokines for damage recovery, but excessive inflammatory reaction may result in tissue damage [2]. Exercise-induced muscle contraction leads to hypertrophic muscle growth and is accompanied by a robust inflammatory response, including the release of ROS and reactive nitrogen species (RNS) [3]. Leukocytes play a central role in the inflammatory response. Neutrophils, which are the most abundant leukocytes in the body, are the first cells to reach a site of inflammation [4]. During the post-exercise acute inflammatory response, neutrophils have several functions: after migration to muscle tissue, they stimulate the production of ROS, RNS, elastase, myeloperoxidase (MPO), and tumor necrosis factor (TNF)-α, resulting in neutrophil extracellular trap (NET) formation, phagocytosis, and necrosis [5,6,7,8,9].

Exercise-induced muscle damage (EIMD) and organ damage are two well-known types of inflammation that can result from exercise. Over the last few decades, a multitude of natural products and compounds with anti-inflammatory and/or antioxidant properties, such as curcumin, tart cherry juice, omega-3 fatty acids, and vitamins C, D, and E, have been tested to alleviate such inflammation [5,10,11]. Although human studies have confirmed the efficacy of food ingredients in reducing exercise-induced inflammation, the underlying mechanism of their action often remains unknown.

Neutrophils play a crucial role in exercise-induced inflammation due to their inflammatory and oxidant effects. Studies in mice have shown that depleting neutrophils and monocytes reduced migration and mitigated the level of the inflammatory cytokines TNF-α and IL-6 in skeletal muscle [7,12]. Therefore, food ingredients that suppress leukocyte function may also be effective in suppressing exercise-induced inflammation.

Sulforaphane (SFN) is an isothiocyanate compound that can potentially reduce exercise-induced inflammation. SFN derives from glucoraphanin, which is present in cruciferous vegetables like broccoli sprouts [13]. Glucoraphanin can be converted to SFN by myrosinase or by assimilation by intestinal bacteria [14]. SFN exhibits both anti-inflammatory and antioxidant properties [15] and may prevent the migration of leukocytes, including neutrophils [16]. Moreover, SFN has been found to attenuate the increase of creatine kinase and interleukin (IL)-6 in humans during heavy resistance exercise [17].

Given that SFN may inhibit leukocyte function during exercise, it is plausible that it could prevent exercise-induced inflammation. However, few studies have examined the effects of SFN on different human neutrophil functions. To address this research gap, we investigated the effectiveness of SFN against several human neutrophil functions using both whole blood and separated neutrophils. Specifically, we evaluated the impact of SFN on ROS production, degranulation, phagocytosis, and NET formation.

## 2. Results

### 2.1. Inhibitory Effect of SFN Treatment on ROS Production of Whole Blood and Neutrophils

Neutrophil function after exercise is evaluated by measuring ROS production and phagocytic activity produced by zymosan stimulation [18,19]. In this study, we followed these methods and measured ROS using a zymosan stimulation model. The current study aimed to investigate the effect of SFN on ROS production in whole blood, neutrophils, and peripheral blood mononuclear cells (PBMCs), respectively. In this method, the cells phagocytosed zymosan to produce ROS. To this end, the relative ROS production was measured using luminol-dependent chemiluminescence. Importantly, SFN did not affect the survival of whole blood-derived neutrophils at any SFN concentration during the ROS measurement period (Figure 1a).

Our results indicated that SFN treatment effectively suppressed ROS production in a concentration-dependent manner (Figure 1b). Specifically, the inhibition of ROS was significant at SFN concentrations above 280 µM, and the area under the curve (AUC) was significantly suppressed at SFN concentrations above 140 µM (Figure 1c,d). 

It is noteworthy that the zymosan-unstimulated condition did not induce ROS production. Furthermore, our findings were consistent across various cell types, including isolated polymorphonuclear neutrophils and peripheral blood mononuclear cells (PBMCs) (Figure 1e,f). Although monocytes are phagocytic and generate ROS reactive with luminol, the luminescence of PBMCs, which includes monocytes, is approximately 1/10th of that of neutrophils. This indicates that the majority of ROS luminescence in whole blood originates from neutrophils. Collectively, these findings suggest that SFN treatment effectively suppresses ROS production in neutrophils in a concentration-dependent manner without affecting their survival.

### 2.2. Exogenous Antioxidant Activity of SFN

#### 2.2.1. Effects of MPO Inhibition

To investigate whether SFN affects the MPO-mediated ROS production pathway, ROS was measured in whole blood using the MPO inhibitor deferoxamine. Luminol and lucigenin were used as photosensitizers. HOCl was mainly detected by luminol and O_2_^−^ and/or H_2_O_2_ were mainly detected by lucigenin [20]. Under the 0 µM SFN condition, deferoxamine significantly attenuated luminescence intensity compared to the luminol-detection (Lm) condition and the luminol-detection plus deferoxamine (Lm + d) condition. (Figure 2a). Interestingly, SFN significantly attenuated luminescence intensity under both the Lm and Lm + d conditions. These results suggest that SFN may suppress MPO and/or remove HOCl.

No effect of SFN was observed under the lucigenin detection (Lg) condition (Figure 2b), whereas luminescence intensity under the lucigenin detection plus deferoxamine (Lg + d) condition was significantly higher than that for the Lg condition. SFN significantly attenuated luminescence intensity under the Lg + d condition. These results suggest that O_2_^−^ and/or H_2_O_2_ accumulate upon the suppression of MPO by deferoxamine. Since SFN attenuated luminescence intensity under the Lg + d condition, it may remove O_2_^−^ and/or H_2_O_2_.

#### 2.2.2. Effects of Exposure Time to SFN

To investigate whether SFN has an exogenous antioxidant effect, we evaluated the impact of exposure time to SFN on ROS production. The exposure time to SFN was set to 0 min, 15 min, and 45 min, followed by stimulation with zymosan. This measurement was performed using whole blood. Chemiluminescence was measured 45 min after stimulation. As shown in Figure 2c, SFN significantly attenuated ROS production regardless of the exposure time, suggesting that SFN can directly remove ROS.

#### 2.2.3. Direct Antioxidation Effect of SFN on HOCl

To investigate whether SFN has a direct antioxidant activity on HOCl, we performed the OXY adsorbent test, a cell-free system that measures the scavenging activity of a compound towards HOCl. Ascorbic acid was used as a positive control, and SFN was compared to it at identical molar concentrations. As shown in Figure 2d, SFN exhibited strong scavenging activity against HOCl. This suggests that SFN has exogenous antioxidant capacity.

### 2.3. Inhibitory Effect on Inflammation and Degranulation

To investigate the effect of SFN on inflammation and degranulation, we collected the sample supernatants after the ROS measurements in whole blood, and measured the concentrations of the inflammatory cytokines TNF-α, IL-1β, and IL-6, as well as the anti-inflammatory cytokine IL-10, and MPO as an indicator of degranulation. As shown in Figure 3, SFN at concentrations of 140 µM and 280 µM significantly decreased the amount of MPO (Figure 3e) and TNF-α (Figure 3a), while only 280 µM SFN significantly decreased IL-6 (Figure 3c). However, the amounts of IL-1β and IL-10 did not differ significantly under any tested condition (Figure 3b,d). These results suggest that SFN has both anti-inflammatory and anti-degranulation properties in whole blood.

### 2.4. SFN Suppresses Neutrophil Phagocytosis

We examined the effect of SFN on phagocytic activity using isolated neutrophils. Figure 4a presents a typical image of phagocytic neutrophils. Both 140 and 280 µM SFN significantly reduced the ratio of phagocytic neutrophils, as well as decreasing the number of phagocytized zymosan particles (Figure 4b,c). These results suggest that SFN inhibits the phagocytic activity of neutrophils.

### 2.5. SFN Does Not Impact NET Formation

To investigate the potential effect of SFN on NET formation, we utilized both the NADPH oxidase-dependent stimulant PMA and the NADPH oxidase-independent stimulant A23187. As shown in Figure 5, NET formation significantly increased under the stimulated condition compared to the non-stimulated condition. However, our results indicate that the presence of SFN did not result in the suppression of NET formation, regardless of the presence of either PMA or A23187 (Figure 5a,b).

## 3. Discussion

The results of this study indicate that SFN significantly decreased the production of reactive oxygen species (ROS) in both whole blood and neutrophils, as well as reducing degranulation and phagocytosis, without affecting neutrophil extracellular trap (NET) formation. Moreover, SFN was observed to directly eliminate ROS when including hypochlorous acid (HOCl).

It is worth noting that neutrophils have a relatively short lifespan of 6–8 h following blood collection, making it necessary to examine them immediately to avoid the effects of cell deterioration. The ex vivo ROS measurement method used in the present study, which employs human whole blood, enables rapid measurement of multiple specimens through the use a plate reader, and is superior to earlier methods. In addition, this method also examines the effects of blood constituents such as lymphocytes, monocytes, erythrocytes, platelets, and other blood cells and plasma proteins on neutrophils. This means that the effects of SFN were tested in an environment that is more similar to an in vivo environment.

Our investigation into the effect of SFN on total neutrophil ROS production yield the observation that SFN attenuated ROS in a concentration-dependent manner. Previous studies have demonstrated that SFN suppresses oxidative stress by interacting with the classic regulator of redox balance, nuclear factor erythroid 2-related factor 2 (Nrf2) [21,22,23,24]. In mice, intraperitoneal administration of SFN reduces neutrophil ROS production [25]. Additionally, in mouse neutrophils and macrophage-like RAW264.7 cells, SFN exposure increases the expression of antioxidant-related genes, such as Nrf2, heme oxygenase (HO)-1, and catalase [21]. Therefore, we initially presumed that the ROS suppression effect of SFN obtained in this study was due to this “endogenous” antioxidant activity of SFN.

However, as shown in the deferoxamine addition experiments, SFN suppressed ROS in whole blood, even when HOCl generation was inhibited by MPO inhibition. This prompted us to further investigate the effects of SFN on stimulated cells before the “endogenous” action, and again SFN inhibited ROS. In addition to its “endogenous” antioxidant activity, we measured the scavenging capacity of HOCl in the cell-free system and found that SFN was capable of scavenging ROS, including at least HCIO, and its antioxidant capacity was higher than that of L-ascorbic acid. Thus, our results suggest that SFN is capable of scavenging ROS “exogenously”. An in silico study has also shown that SFN is capable of scavenging O_2_^−^, H_2_O_2_, and HOCl in the presence of Fe-superoxide dismutase (SOD) [26], which is consistent with our results. Based on the above results, SFN is expected to exert a strong antioxidant effect in vivo through both “endogenous” and “exogenous” antioxidant effects.

In addition to measuring ROS levels, we also measured inflammation-associated proteins from the supernatant of the ROS detection assay using whole blood. We found that zymosan stimulation significantly increased the inflammatory cytokine TNF-α, suggesting that inflammation was induced in the wells. Furthermore, the addition of SFN suppressed TNF-α and IL-6 levels. These findings are generally consistent with the results of a previous study that used RAW264.7 cells [21]. Thus, our results indicate that SFN has anti-inflammatory effects, even in whole blood.

MPO was also measured in the same way as inflammatory and anti-inflammatory cytokines. Our result showed that SFN inhibited the release of MPO, suggesting that SFN inhibits degranulation. Based on the ratio of neutrophils to leukocytes and the expression of MPO, the measured MPO mainly originates from neutrophil. Neutrophil degranulation is caused by the activation of surface proteins such as selectins and integrins, which activate a kinase cascade [27,28]. However, there are few reports on the inhibitory effect of SFN on neutrophils and on degranulation, and further investigation is needed. Analysis of these adhesion factors and their cascades will also be required to fully understand the mechanism of action of SFN on neutrophils.

In this study, our results indicate that SFN also decreased phagocytosis. However, a previous study using Nrf2-knockout polymorphonuclear neutrophils reported that Nrf2 did not have an effect on phagocytosis [29]. This suggests that Nrf2 may not be involved in phagocytosis. Interestingly, several studies have demonstrated that SFN impacts leukocyte adhesion [30,31]. Based on these observations, it is possible that SFN regulates neutrophil adhesion factors, and thereby inhibits phagocytosis and degranulation by altering adhesion factors.

Unexpectedly, SFN did not affect NET formation in our study. ROS is involved in chromatin aggregation in the early stage of NET formation [32]. Lack of NADPH oxidase function causes a decrease in ROS and NET formation [33]. Moreover, antioxidant substances such as vitamin C and N-acetyl cysteine (NAC) suppress NET formation [34,35]. On the other hand, the suppression of NETs by Vitamin C and NAC has been tested at concentrations of 1 mM or higher. In this study, NETs induction was tested at lower concentrations, where endogenous antioxidant capacity has been reported, because of the long duration of the study. From these results, it is expected that treatment with a higher concentration is necessary for SFN to inhibit NETs formation. However, that concentration is not realistic in SFN.

It is worth noting that our study employed short-duration measurements with high concentrations of SFN, which differs from in vivo conditions. The concentration of SFN in blood when SFN is taken orally typically reaches several hundred nmol [36,37]. Experiments conducted using lower concentrations and for longer durations (while still compatible with neutrophil survival) would provide results more comparable with those observed in vivo. In addition, further analysis with isolated neutrophils is necessary, as tests using whole blood are affected by blood cells and other components of blood.

It was previously reported that administration of SFN inhibits leukocyte infiltration into tissues [25,30]. This effect may be attributed to the antioxidant and anti-inflammatory properties of SFN, as well as its ability to suppress the expression of vascular endothelial cell adhesion factors, thus suppressing infiltration. Given that SFN can act on a variety of cells, including platelets, monocytes, and neutrophils, it is expected to have a broad range of anti-inflammation and anti-oxidative effects. Lymphocyte and monocyte counts increase immediately after exercise, with a delayed increase in neutrophils [38]. Further studies are expected to determine how SFN in the blood affects leukocyte counts after exercise.

This study showed that SFN has antioxidant activity on human whole blood containing neutrophils. When exercise causes damage to organs and muscles, neutrophils and macrophages infiltrate the tissue and induce inflammation. Taking SFN before exercise may suppress the activity of blood cells, including neutrophils, and may reduce inflammation. However, despite the important role of neutrophils in exercise-induced inflammation, studies investigating their function are limited and details about their function remain largely unknown. Therefore, the findings from this study may also contribute to a better understanding of neutrophil function and facilitate future research in this area.

## 4. Materials and Methods

### 4.1. Handling of Blood Samples

Peripheral blood samples were obtained from a cohort of seven healthy male and female volunteers between the ages of 20 and 60 years. The blood samples were obtained via venipuncture and collected in Vacutainers containing heparin. The blood samples were used within two hours, during which they were kept at room temperature. A portion of each blood sample was used as whole blood, while another portion was used for the detection of cell-specific ROS following the isolation of neutrophils and PBMCs.

### 4.2. Cell Isolation

Neutrophils and PBMCs were isolated from whole blood using the double-density gradient centrifugation method, as modified from a previous study [20]. Heparinized venous blood was layered onto equal volumes of Histopaque-1119 (Sigma Aldrich, St. Louis, MO, USA) and Histopaque-1077 (Sigma Aldrich, St. Louis, MO, USA), then centrifuged at 400× *g* for 30 min at room temperature. The neutrophil fraction is located at the 1077–1119 interface, and was collected separately, and washed twice with HBSS. The number of cells and quality of purification (>90% in leukocytes) were determined using an automatic blood cell counter (pocH-100i, Systemic, Hyogo, Japan) capable of distinguishing between neutrophils, monocytes, and lymphocytes. Experiments with isolated cells were performed with *n* = 6 due to the low neutrophil counts in the subjects’ blood.

### 4.3. Detection of ROS by Chemiluminescence

Detection of ROS was performed using a chemiluminescence assay based on a previous published method [20]. Relative ROS production levels were detected using photosensitizers. The photosensitizers are excited by oxygen species released from cells that have phagocytosed zymosan.

The assay was performed using a 96-well microplate, and solutes were dissolved with HBSS (ThermoFisher Scientific, Rockford, IL, USA). Luminol (0.875 mM) (A-8511, Sigma Aldrich, St. Louis, MO, USA) or 0.5 mM lucigenin (M-8010, Sigma Aldrich, St. Louis, MO, USA), 0 to 280 µM SFN (S8044, LKT Laboratories, St. Paul, MN, USA), and 35 µL whole blood or isolated cells (8 × 10^4^ cells/well for neutrophils and 1.5 × 10^5^ cells/well for PBMCs), and/or 15.2 mM deferoxamine (Deferoxamine Mesilate; Novartis Pharma, Tokyo, Japan) were added to each well. Each sample was mixed thoroughly, incubated, and the luminescence was measured at 37 °C, for 45 min. Following the incubation period, 10 µL of 1 mg/mL zymosan A (Z-4250, Sigma Aldrich, St. Louis, MO, USA) was then added to each well and the samples were continuously measured at 37 °C, for an additional 45 min. The final sample volume in each well was 100 µL. A microplate reader (SpectraMax iD5, Molecular Devices, LLC., San Jose, CA, USA) was used to measure chemiluminescence. Measurements were taken every 5 min and samples were shaken before each measurement. All samples were measured in duplicate.

### 4.4. Cell Viability Assay

Cell viability was evaluated using the Cell Counting Kit-8 (CCK-8) Kit (Dojindo, Kumamoto, Japan) according to the manufacturer’s instructions. Isolated neutrophils and PBMCs at a seeding density of 1.5 × 10^4^ cells/well were plated in 96-well plates. SFN was then added to each well and the plates were incubated at 37 °C, under 5% CO_2_ for 90 min. Following the incubation period, the supernatant was carefully removed, and the reaction reagent from the CCK-8 kit was added. The samples were incubated at 37 °C, under 5% CO_2_, for an additional 90 min. All samples were analyzed in duplicate. 

### 4.5. OXY Adsorbent Test

The OXY adsorbent test involves exposing a sample to a solution of hypochlorous acid (HOCl) with a known oxidation capacity. Antioxidants in the sample react with the acid, and the antioxidant capacity is quantified by measuring the excess HOCl remaining in the solution. After a specified reaction time, the residual HOCl is reacted with a chromogenic agent, such as an alkyl-substituted aromatic amine, which results in a pink color.

A commercially available OXY adsorbent test kit (Wismer, Tokyo, Japan) was used as the measuring reagent. The absorbance was measured using a specialized instrument, the Free Radical Elective Evaluator (Wismer). The procedure was performed according to the manufacturer’s instructions. Briefly, 10 µL of the sample was mixed with 1 mL of HClO solution, and kept warm at 37 °C for 10 min. Following this, a drop of chromogenic solution chromogen was added, and the sample was measured at 546 nm for 3 s. The HOCl concentration was determined by comparing the sample absorbance to that of a blank solution. The eliminated HClO concentration was recorded as the results in µmol HClO/mL.

SFN and L-ascorbic acid (012-04802; Fujifilm-Wako, Osaka, Japan) were diluted with distilled water and adjusted to the same molar concentration. The measurement was repeated three times for each sample.

### 4.6. Enzyme-Linked Immunosorbent Assay (ELISA)

The supernatants after ROS measurements using whole blood were collected and used as samples. The concentrations of cytokine and other inflammation-related proteins were measured in the supernatant samples using enzyme-linked immunosorbent assay (ELISA) kits according to the manufacturer’s instructions. The concentrations of TNF-α, IL-1β, and IL-6 were measured using Quantikine^®^ kits (R&D Systems, Inc., Minneapolis, MN, USA). IL-10 was measured with a BD OptEIA™ Human IL-10 ELISA Kit II (Becton, Dickinson and Company, Franklin Lakes, NJ, USA). Myeloperoxidase (MPO), a component of neutrophil granules, was measured with an MPO Human Kit (Hycult Biotech, Uden, The Netherlands). In all assays, absorbance was spectrophotometrically measured with the SpectraMax iD5 microplate reader. The concentration of each protein in each sample was calculated from a standard curve established using the same measurement conditions.

### 4.7. Phagocytosis Assay

Isolated neutrophils were seeded at a density of 5 × 10^4^/well in a 1.5 mL tube. This was followed by the addition of SFN and incubation at 37 °C with 5% CO_2_ for one hour. The final SFN concentrations tested were 0, 140, and 280 µM. The cells were then stimulated with opsonized zymosan and incubated at room temperature for 25 min. The preparation of opsonized zymosan was according to the method established in a previous study [21]. The reacted cell suspensions were gently pipetted, and 20 µL of the cell suspension was dropped onto glass slides and rapidly air-dried to fix the cells. These samples were then stained with May–Grunwald Giemsa staining and examined at 400× magnification using a bright-field microscope BH-2 (Olympus, Tokyo, Japan). Approximately 100 neutrophil cells were counted on each slide, and phagocytic cells were identified based on the following criteria: (α) the cells must not overlap with other cells, (β) the cells must have particles inside them, and (γ) the cells must have a closed cell membrane.

### 4.8. NET Formation Assay

The NET formation assay was performed as previously described [39]. Isolated neutrophils were cultured in RPMI1640 (ThermoFisher Scientific) with 5% fetal bovine serum (FBS), and treated with 0, 10, and 20 µM of SFN at 37 °C under 5% CO_2_, for 1 h. NET formation was induced by 1 µM PMA (Sigma-Aldrich, St. Louis, MO, USA) and 1 µM A23187 (Sigma-Aldrich, St. Louis, MO, USA). NET formation was detected using 5 µM SYTOX^®^ Green (ThermoFisher Scientific, Waltham, MA, USA) and monitored using the SpectraMax iD5 microplate reader with excitation at 488 nm and emission at 510 nm. The fluorescence was measured at 37 °C for 4 h, with measurements taken every 5 min and the samples were shaken before each measurement. The peak height of the fluorescence was used for analysis. All samples were assayed in duplicate.

### 4.9. Statistical Analysis

Data are expressed as mean ± standard error (SE). All data analysis was performed using IBM SPSS v27 (IBM Japan, Ltd., Tokyo, Japan). To calculate statistical significance, one-way ANOVA was used for the neutrophil activity, ELISA, phagocytosis assay, and NET formation assay. For the deferoxamine-containing ROS assay, two-way ANOVA was performed, coupled with Tukey’s post hoc test. The level of statistical significance was set to *p* < 0.05.

## Figures and Tables

**Figure 1 ijms-24-08479-f001:**
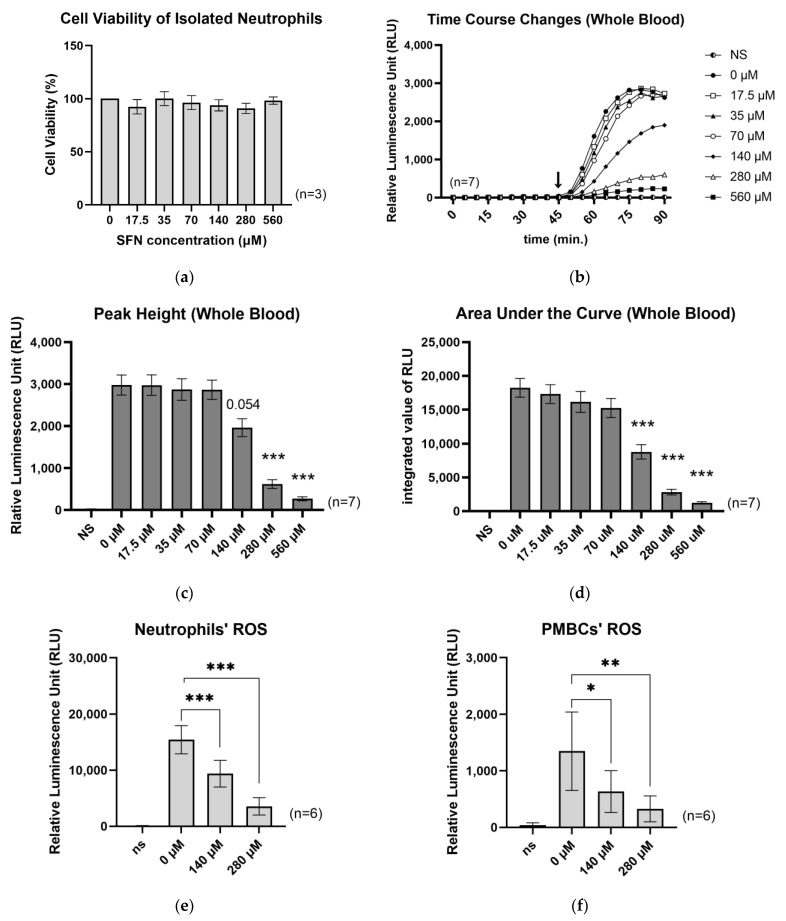
Effects of SFN on neutrophil viability and ROS production. (**a**) Cell viability of isolated neutrophils. (**b**) Time course changes in whole blood ROS production. (**c**) Peak height of ROS production, from the data shown in Figure 1a, (**d**) Area under the curve (AUC) of ROS production, from the data shown in Figure 1a, (**e**) ROS production using isolated neutrophils, (**f**) ROS production using PBMCs. The arrow in the Figure 1b indicates the time of stimulation by zymosan. Values are the mean and S.E. (*n* = 3 for Figure 1a, *n* = 7 for Figure 1b–d, *n* = 6 for Figure 1e,f). NS, no stimulation by zymosan. Each concentration in the X axes indicates SFN concentration. * *p* < 0.05, ** *p* < 0.01, *** *p* < 0.001, significantly different compared with 0 µM SFN.

**Figure 2 ijms-24-08479-f002:**
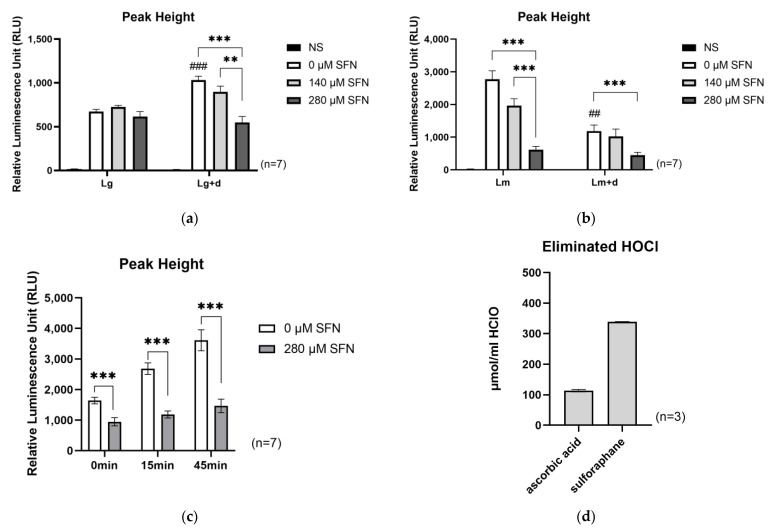
Direct removal of ROS by SFN using whole blood and cell-free system. (**a**) ROS measurement by luminol and MPO suppression by deferoxamine. Lm, detected using luminol; Lm + d, detected using luminol and deferoxamine. (**b**) ROS measurement using lucigenin and MPO suppression by deferoxamine. Lg, detected using lucigenin; Lg + d, detected using lucigenin and deferoxamine. (**c**) Time-independent ROS suppression activity of SFN. The X axis indicates the incubation time with SFN. (**d**) Cell-free OXY adsorbent test using 5.6 µM of L-ascorbic acid or SFN. Values are the mean and S.E. (*n* = 7 for Figure 2a–c, *n* = 3 for Figure 2d). ## *p* < 0.01, ### *p* < 0.001, significantly different compared with 0 µM SFN without deferoxamine. ** *p* < 0.01, *** *p* < 0.001, significantly different between indicated groups.

**Figure 3 ijms-24-08479-f003:**
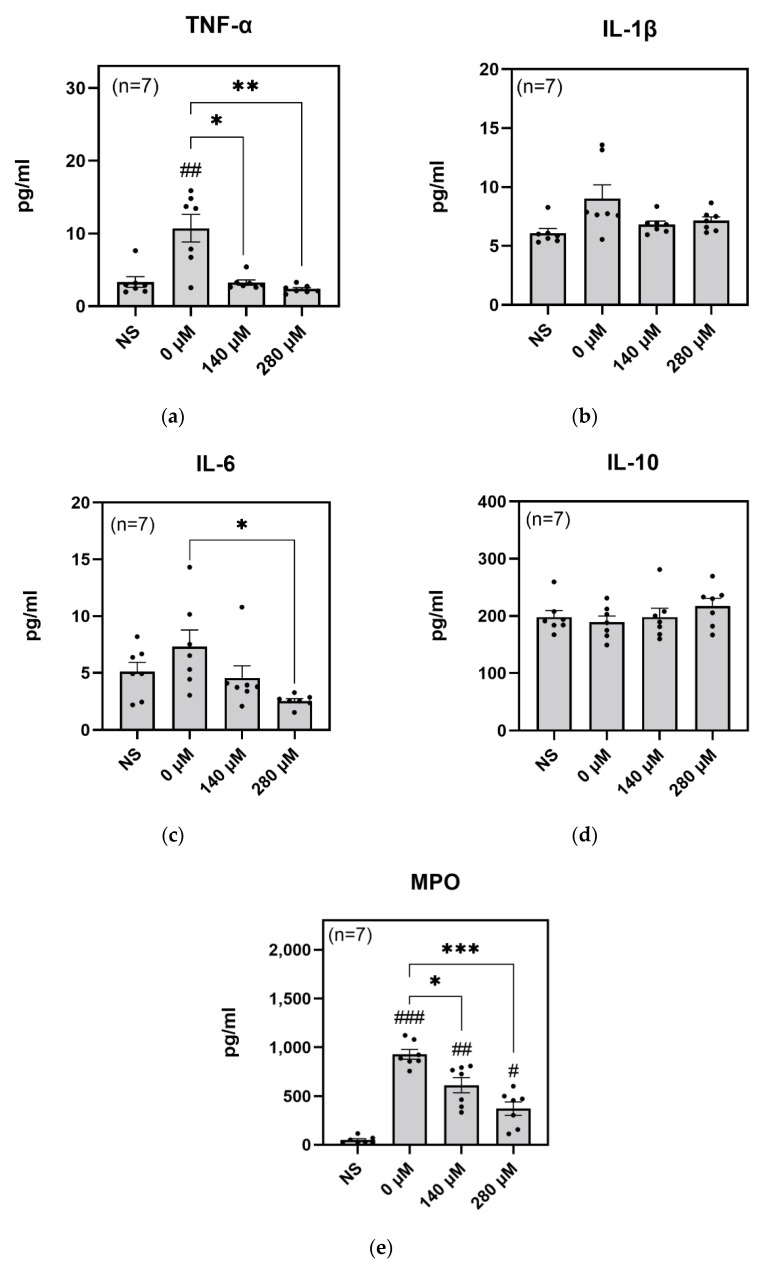
Amounts of various proteins in the reaction supernatants from the whole blood ROS measurement. The concentrations of the inflammatory cytokines (**a**) TNF-α, (**b**) IL-1β, and (**c**) IL-6, that of (**d**), the anti-inflammatory cytokine IL-10, and (**e**), the MPO content of azure granules, were detected by ELISA. NS indicates no stimulation by zymosan. The X axes indicate SFN concentration. Values are the mean and S.E. (*n* = 7). # *p* < 0.05, ## *p* < 0.01, ### *p* < 0.01, significantly different compared with NS. * *p* < 0.05, ** *p* < 0.01, *** *p* < 0.001, significantly different compared with 0 µM SFN.

**Figure 4 ijms-24-08479-f004:**
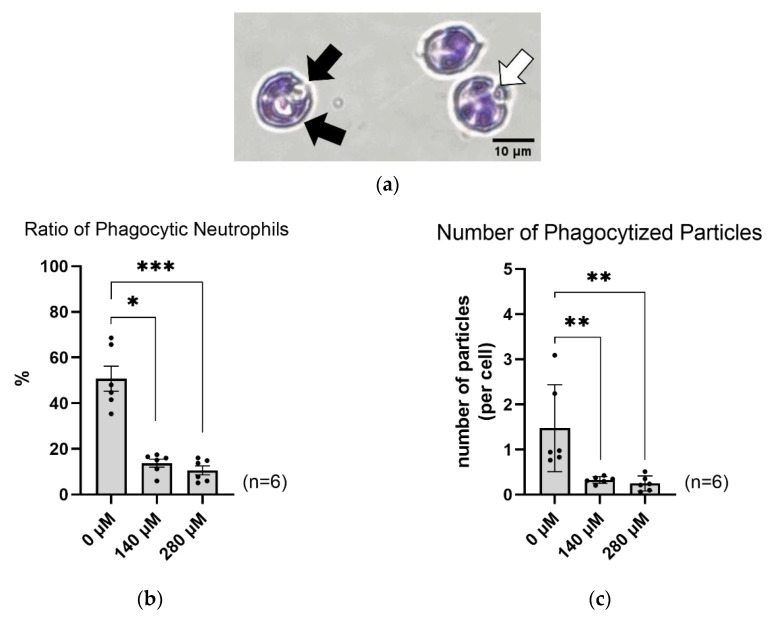
Effect of SFN on the phagocytic activity of isolated neutrophils. (**a**) A typical image of phagocytic neutrophils, (**b**) Percentage of phagocytized neutrophils out of the total measured neutrophils. (**c**) The number of phagocytized opsonized zymosan particles. The black arrows indicate phagocytized zymosan particles. The white arrow indicates an un-phagocytized zymosan particle. Cells that have purple polymorphonuclears are neutrophils. The smaller cells are red blood cells. The X axes indicate SFN concentration. The values are the mean and S.E. (*n* = 6). * *p* < 0.05, ** *p*<0.01, *** *p* < 0.001, significantly different compared with 0 µM SFN.

**Figure 5 ijms-24-08479-f005:**
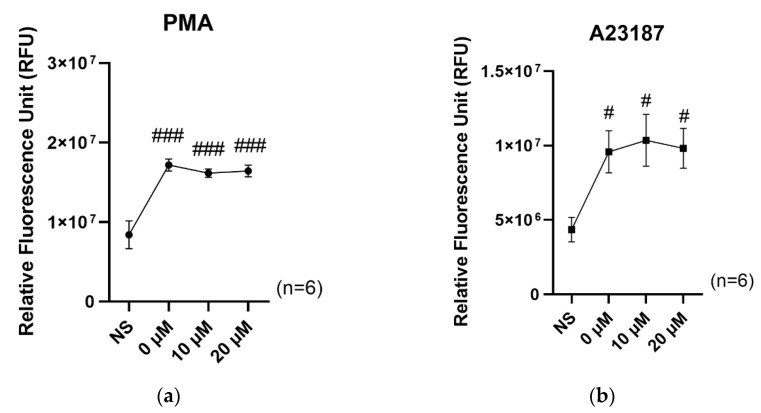
Effect of SFN on NET formation by neutrophils, stimulated by (**a**) 1 µM PMA and (**b**) 1 µM A23187. NS indicates no stimulation. The X axes indicate SFN concentration. The values are the mean and S.E. (*n* = 6). # *p* < 0.05, ### *p* < 0.001, significantly different compared with NS.

## Data Availability

Data are contained within the article.

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
