# Peer review of "Sulforaphane Attenuates Neutrophil ROS Production, MPO Degranulation and Phagocytosis, but Does Not Affect NET Formation Ex Vivo and In Vitro"

_ijms, 2023, doi:10.3390/ijms24108479_

Round 1
Reviewer 1 Report (New Reviewer)
The results are relevant the use of natural antioxidants in the control of neutrophil ROS could be useful in some physiological and pathological conditions however the author try to do a relationship with inflammation in the exercise and fail to justify the zymosan model
To do a comparation between antioxidants is necessary to add the redox potential value including Vit C, N acetyl cysteine and Glutathione that are THIOLS like SFN and are present in the cells.
In the exercise are changes in the leukocytes proportion this is important because authors are focusing in inflammation by neutrophils add to discussion more information about this.
Vit C reduce Nets formation moreover N-acetyl cysteine "NAC" a thiol like SFN has been used to reduced NETs formation then why SFN fail to reduce NETS is a contradictory results and must be discussed
How is the mechanism that used the SFN to go inside the cells ?
Phagocytosis, ROS and HCLO are necessary to eliminate pathogens and after NETs continue killing but the neutrophil die then using SFN means the capacity of the neutrophils to eliminate microorganisms are reduced and moreover go directly to NETs reducing the amount of neutrophil?
Author Response
We extend our heartfelt gratitude to you for taking the time to review our manuscript titled " Sulforaphane Attenuates Neutrophil ROS Production, MPO Degranulation and Phagocytosis, but Does Not Affect NET Formation: ex vivo and in vitro". We appreciate your valuable comments and suggestions. We have carefully considered the points raised in your review and would like to respond to them as follows.

Reviewer 2 Report (New Reviewer)
Dr Wakasugi-Onogi and colleagues present an interesting draft describing the effect of sulforaphane on the bactericidal activity of neutrophils.
The ever-increasing number of publications on Sulforaphane and its biological effects highlights the importance and relevance of this study.
After reviewing the paper, I found a few crucial things that must be fixed during the improvement the draft.
1) how do the authors justify the choice of SFN concentrations in the experiments? Is it possible to use SFN at such concentrations in vivo?
2) Please explain why SFN was used at different concentrations to determine the formation of NETs (10 µM and 20 µM) and in tests for phagocytosis and determination of ROS and cytokine production, OXY adsorbent test (140 µM and 280 µM)? The authors write "Unexpectedly, SFN did not appear to affect NET formation in our study". It appears that SNF had no effect on NETs formation due to the use of concentrations that did not cause suppression of ROS production (Fig. 1b). Based on the results obtained, it is not possible to conclude that SFN affects neutrophil ROS production, MPO degranulation and phagocytosis, but does not affect NET formation, as indicated in the title and conclusions because the effect has been studied for different concentrations of the active substance (SFN).
3) The authors say "SFN did not affect the survival of whole blood-derived neutrophils at any SFN concentration during the ROS measurement period (Figure 1a)", but no statistical comparison was made between treatment groups. Why then do the authors make such a conclusion?
Minor
1) Scale bar should be added to the micrograph (Figure 4a). In general, the quality of the photomicrograph is very low, I recommend the authors to use other methods of fixation (ethanol, PFA etc.) and immersion lens – it will greatly simplify the analysis of microscopic preparations.
Author Response
We extend our heartfelt gratitude to you for taking the time to review our manuscript titled " Sulforaphane Attenuates Neutrophil ROS Production, MPO Degranulation and Phagocytosis, but Does Not Affect NET Formation: ex vivo and in vitro". We appreciate your valuable comments and suggestions. We have carefully considered the points raised in your review and would like to respond to them as follows.

Round 2
Reviewer 2 Report (New Reviewer)
The authors have answered some of the reviewer’s criticisms and the manuscript is now more accurately presented.
This manuscript is a resubmission of an earlier submission. The following is a list of the peer review reports and author responses from that submission.
Round 1
Reviewer 1 Report
I would like to thank editors for permitting me to review the manuscript “Sulforaphane Attenuates Neutrophil ROS Production, MPO Degranulation and Phagocytosis, but Does Not Affect NET Formation” by Shiori Wakasugi-Onogi1, Sihui Ma, Ruheea Taskin Ruhee, Yishan Tong, Yasuhiro Seki and Katsuhiko Suzuki. The article is devoted to the study of the effect of sulforaphane, a natural substance present in cruciferous vegetables and having anti-inflammatory properties, on the function of neutrophils, which play an important and dual role in inflammatory processes. The main questions to the authors of the article arose in connection with the too brief and insufficient presentation of the methods and conditions of the experiments. In addition, the paradoxical result of the study of phagocytosis and the use of SY TOX Green as a marker of NETs have raised objections.
СOMMENTS
Figure 1.
In abstract, lines17-18
“First, we measured neutrophil ROS production induced by zymosan in whole blood in the presence of 0 to 560 μM sulforaphane”
In Methods, Neutrophil activating assay, ROS production, line 240
“and 35 μl whole blood were added to each well”.
In Results:
“2.1. Inhibitory Effect of SFN Treatment on ROS Production of Neutrophils”.
In the text of this paragraph and in the legend to Figure 1, the authors mention "neutrophils" everywhere. It remains unclear whether the authors performed their experiments with whole blood or with isolated neutrophils. The difference between the results of experiments on whole blood and isolated neutrophils can be huge. Other blood cells, such as monocytes, and serum proteins also influence the results.
Authors should indicate how the zymosan particles were prepared for the experiments and how much zymosan was added to each well.
Lines 76-77: “SFN did not affect the survival of whole blood-derived neutrophils at any SFN concentration tested (data not shown)”. Authors should describe how they determined neutrophil survival, especially if they use whole blood.
1. Figure 2.
Line 118: “Direct removal of ROS by SFN. (a) ROS measurement by luminol and MPO suppression by deferoxamine.” I did not understand these experiments and did not find this technique in the Methods. What was the source of ROS in these experiments?
2. Figure 3 Authors again did not indicate the experimental conditions.
Lines 127: “We collected the sample supernatants after the ROS measurements and measured the concentrations of the inflammatory cytokines”.
Was ROS production studied on isolated neutrophils or whole blood? Lyphocytes and monocytes of whole blood can greatly alter the release of cytokines.
3. Figure 4
The title in the figure: “Number of phagocytosed opZymosan”. I would guess that this is the average amount of zymosan particles ingested by one neutrophil. But in this case, Fig. 1 shows that one neutrophil ingested 150 particles at 0 μM of the test preparation.
A neutrophil is absolutely unable to phagocytose 150 particles under any circumstances! But this result may indicate strong aggregation of zymosan particles. The authors should indicate the method of preparation of zymosan particles for experiments, whether the particles were opsonized or not, the ratio of the number of particles to the number of neutrophils in the experiments. It will be very helpful to show a photograph of a typical phagocytosis scoring slide. It can clearly demonstrate the status of neutrophils and zymosan particles. In Methods phagocytosis assay and slide preparation must be exactly and clear described.
A mistake in the word: “nutrophils”
4. Figure 5
SY TOX Green is a specific stain for nucleic acids that penetrates dead cells but does not penetrate living cells. An increase in the fluorescence intensity of this dye may indicate an increase in the number of dead cells. Why do the authors believe that the fluorescence of this dye indicates the formation of NET? What structural or functional specificity of NET can be revealed by the intensity of SY TOX Green?
In the methods, the authors indicated that they used test drug concentrations (SEN) of 10 and 20 µM, and in Figure 5, test drug concentrations of 140 and 280 µM are indicated.
The concentration of PMA and A23187 is not indicated at all.
Author Response
Thank you for inviting us to submit a revised draft of our manuscript entitled, “Sulforaphane Attenuates Neutrophil ROS Production, MPO Degranulation and Phagocytosis, but Does Not Affect NET Formation” to International Journal of Molecular Sciences. We also appreciate the time and effort you have dedicated to providing insightful feedback on ways to strengthen our paper. Thus, it is with great pleasure that we resubmit our article for further consideration. We have incorporated changes that reflect the detailed suggestions you have graciously provided. We also hope that our edits and the responses we provide below satisfactorily address all the issues and concerns you and the reviewers have noted.

Reviewer 2 Report
In the manuscript, “Sulforaphane Attenuates Neutrophil ROD Production, MPO Degranulation and Phagocytosis, but does not Affect NET Formation,” the authors examined the effect of sulforaphane, an isothiocyanate compound, on several aspects of neutrophil function. Ultimately, the goal of the authors is to develop potential therapeutics for exercise-induced inflammation and tissue damage. The authors present convincing data that SFN has an effect on ROS production, phagocytosis and cytokine production but not NETosis in an ex vivo model system. There are a few major points that detract from the manuscript. The use of whole blood vs isolated neutrophils should be clarified in the manuscript and the use of whole blood be further justified where it is used. The authors present strong evidence that SFN acts through Nrf2 signaling, but I was disappointed that experiments directly testing this link was not tested in their system. Addition of these experiments would greatly enhance the manuscript. Overall, this is a clearly presented manuscript presenting ex vivo data showing a link between SFN and some neutrophil functions. It is an early steppingstone in elucidating the overall pathway, and potential therapeutic uses of SFN. Additional details regarding the above concerns are detailed below, as well as a few other minor points to correct/clarify.
It is not clear in the manuscript whether the authors used whole blood or isolated/enriched neutrophils for the experiments presented in the study. From the methods section (neutrophil activating assay), it seems that luminol and lucigenin assays were performed on whole blood. In the figure legends for Fig 1 it indicates neutrophils, and in figure 2 cell type (whole blood vs neutrophils) is not indicated. In the discussion you also explain that you had superior results using whole blood. Please provide additional explanation of when you used whole blood versus neutrophils. For those experiments using whole blood, please explain your rationale for interpreting that the effects seen are the results of neutrophil actions only. For example, monocytes and macrophages can also produce ROS and cytokines.
Methods outlining the OXY-absorbent test (Fig 2d) are not included in the methods section. Please add.
Please add the “N” for each experiment to either the individual figure legends or to the Statistical Analysis section of the methods. Please also indicate the number of technical and biologic replicates for each experiment.
Please add to methods section for Phagocytosis assay the definitions used for “phagocytic neutrophils” how opsonized zymosan particles were quantified, and how many slides were counted per condition.
Please provide additional explanation as to how Sytox Green was used to quantify NET formation. Or provide a reference where this is explained in detail.
In the discussion section, the authors hypothesize that SFN acts through Nrf2 signaling. It would add a great deal to the manuscript to test that hypothesis in this system by measuring the effect of SFN on Nrf2 or by blocking Nrf2 signaling and see whether this abrogates the effect of SFN on neutrophils.
Minor comments
Typo in legend for Figure 3e “the MPP content” should be “the MPO content”
Typo in figure titles for Figure 4 “Nutrophils” should be “Neutrophils”
Author Response

(The authors gave the same response as above.)

Round 2
Reviewer 1 Report
I read the answers to my comments of the authors of the article “Sulforaphane Attenuates Neutrophil ROS Production, MPO Degranulation and Phagocytosis, but Does Not Affect NET Formation” by Shiori Wakasugi-Onogi1, Sihui Ma, Ruheea Taskin Ruhee, Yishan Tong, Yasuhiro Seki and Katsuhiko Suzuki .
In response to my questions, the authors specified (in the text and in the methods, in the figures and in the captions to the figures) that the vast majority of the experiments presented in Figures 1, 2, and 3 were performed not on isolated neutrophils, but on whole blood. Isolation of neutrophils from the blood requires a sufficient amount of time and good qualifications of experimenters. It is much faster and easier to use a drop of whole blood. But as you know: "The number is bigger, the price is cheaper." The contribution of neutrophil activities to whole blood data is difficult to assess. Other blood cells (platelets, erythrocytes, lymphocytes, monocytes and so on), serum proteins, and even heparin added to prevent blood clotting also contribute to the results. Experiments of this level, in my opinion, do not correspond to a journal with a high impact factor.
The main questions to the authors of the article arose in connection the paradoxical result of the study of phagocytosis. Figure 4 of the old variant of the article showed that one isolated neutrophil ingested 150 particles in the absence and approximately 30 particles in the presence of sulforaphane. In revised version of the article one neutrophil ingested 1,5 particles in the absence and 0,5 particles in the presence of sulforaphane. That is, after the revision, the number of phagocytosed particles changed 100 times. Such paradoxical changes in the results are all the more surprising, since both cells and particles of zymosan are clearly visible under a conventional microscope.
I encouraged the authors to submit a photograph of a typical preparation used to enumerate particles phagocytosed by isolated neutrophils. In the picture placed in Figure 4, 4 neutrophils and 8 erythrocytes are shown in the field of view. That is, neutrophils make up no more than 50% of the cell population. The standard procedure for isolation of neutrophils involves the lysis of red blood cells. Typically, neutrophils make up 96 - 98% of the total number of cells in the final preparation. Therefore, the picture provided by the authors suggests that the study of phagocytosis (Figure 4) was also carried out using a drop of whole blood.
Authors did not answer why do they believe that the fluorescence of SY TOX Green indicates the formation of NETs? What structural or functional specificity of NETs can be revealed by the intensity of SY TOX Green?

Author Response
Re: "Sulforaphane Attenuates Neutrophil ROS Production, MPO Degranulation and Phagocytosis, but Does Not Affect NET Formation” by Shiori Wakasugi-Onogi1, Sihui Ma, Ruheea Taskin Ruhee, Yishan Tong, Yasuhiro Seki and Katsuhiko Suzuki .
Thank you for your reply dated March. 4. We were pleased to learn that our manuscript "Sulforaphane Attenuates Neutrophil ROS Production, MPO Degranulation and Phagocytosis, but Does Not Affect NET Formation” was evaluated as being potentially acceptable for publication in the International Journal of Molecular Sciences, depending on adequate revision and response to the comments raised by the referees.
While we agree with almost all the comments and suggestions made by the two referees, we do not agree with comments 1 and 3. Our detailed replies, concerning both these comments, are contained in our point-by-point responses to the comments.

Reviewer 2 Report
All concerns have been sufficiently addressed by the authors in the revised manuscript.
Author Response
Thank you for your acceptance
Re: "Sulforaphane Attenuates Neutrophil ROS Production, MPO Degranulation and Phagocytosis, but Does Not Affect NET Formation” by Shiori Wakasugi-Onogi1, Sihui Ma, Ruheea Taskin Ruhee, Yishan Tong, Yasuhiro Seki and Katsuhiko Suzuki .
Dear Reviewer #2
Thank you for your letter dated March. 4. We were pleased to learn that our manuscript "Sulforaphane Attenuates Neutrophil ROS Production, MPO Degranulation and Phagocytosis, but Does Not Affect NET Formation” was evaluated as being potentially acceptable for publication in the International Journal of Molecular Sciences, depending on adequate revision and response to the comments raised by the referees.
We thank you for your time and valuable comments and suggestions, which helped us to improve the quality of our manuscript significantly. We hope that this article will be published.
Sincerely Yours,
Round 3
Reviewer 1 Report
I got answers to round 2 of my comments of the authors of the article “Sulforaphane Attenuates Neutrophil ROS Production, MPO Degranulation and Phagocytosis, but Does Not Affect NET Formation” by Shiori Wakasugi-Onogi1, Sihui Ma, Ruheea Taskin Ruhee, Yishan Tong, Yasuhiro Seki and Katsuhiko Suzuki .
In the course of the discussion, the authors of the article and I found out that practically all experiments were performed with whole blood but not with isolated neutrophils. In the last reply, the authors present this fact as an advantage of their work: “As you know, neutrophil works together with other factors in the body, and it is more in the physiological conditions. In this method, neutrophil activity can be measured without the alteration of neutrophil functions by separating process in the conventional methods”.
In this case, the authors are obliged to rename the article in full accordance with the content, for example " Sulforaphane Attenuates ROS Production, MPO Degranulation and Phagocytosis by Whole Blood……”. They have to rewrite the text and discussion, where they can discuss the contribution of neutrophils to the data. But they also must explain why they exclude the contribution of platelets, monocytes, erythrocytes and other blood cells and serum proteins to data. They must also not forget that blood cells and serum proteins can influence the neutrophil response through various metabolic pathways.
Authors did not answer for my comment: “In the picture placed in Figure 4, 4 neutrophils and 8 erythrocytes are shown in the field of view. That is, neutrophils make up no more than 50% of the cell population. The standard procedure for isolation of neutrophils involves the lysis of red blood cells. Typically, neutrophils make up 96 - 98% of the total number of cells in the final preparation. Therefore, the picture provided by the authors suggests that the study of phagocytosis (Figure 4) was also carried out using a drop of whole blood”.
I want to hope that the editors got a tif file of this photo placed in Figure 4, which is very similar to the drawing., which is very similar to the drawing.
Authors did not answer why they believe that the fluorescence of SYTOX Green indicates the formation of NETs. How the authors, using their method, distinguish the remains of destroyed cells containing, among others, DNA molecules from "NETs". What is NETs?
Author Response
Thank you for your time and effort in reviewing our manuscript. We appreciate your insightful comments and suggestions, which have helped us to improve the quality of our manuscript. We have carefully considered your feedback and have made the necessary revisions to address your concerns.

Round 4
Reviewer 1 Report
I got answers to my comments of the authors of the article “Sulforaphane Attenuates Neutrophil ROS Production, MPO Degranulation and Phagocytosis, but Does Not Affect NET Formation” by Shiori Wakasugi-Onogi1, Sihui Ma, Ruheea Taskin Ruhee, Yishan Tong, Yasuhiro Seki and Katsuhiko Suzuki .
In the course of the discussion, the authors of the article and I found out that practically all experiments were performed with whole blood but not with isolated neutrophils.
I suggested renaming the article in full accordance with the content that is address the results to whole blood studies. The authors did not rename the article. In the text of the article it is difficult, and sometimes impossible, to understand how the experiments were performed - on whole blood or neutrophils. The authors interpret the results as neutrophil activity without regard to the contribution of other blood cells and serum components. It is not sufficient to write "we believe" in the discussion, but it should be experimentally proven, that other blood cells and serum components do not affect the results.
The authors claim that they conducted experiments on the study of phagocytosis on isolated neutrophils using light microscopy. From the photograph, it is easy to determine the purity of the neutrophil fraction, as well as the condition of the cells. I asked the authors to add to Figure 4 a photograph of one of the typical preparations they used to quantify phagocytosis. Instead, the authors posted a homemade drawing in Figure 4. The drawing demonstrates the author's imagination, and nothing more.
I believe that data obtained from whole blood samples cannot be attributed to neutrophils. Such work does not correspond to the current level of knowledge in this area. It is even more outdated to draw microscopic images of neutrophils by hand.